


**Review article: Re-viewing Berlin's Urban Parks from the Perspectives of Socio-Economic**
**Inequality, Climate Resilience, and Sustainable Management**
Subham Mukherjee[1], Kei Namba[2], Katrin M. Nissen[3], Ehsan Razipoor[1], Stefan Heiland[2], Brigitta
Schütt[1]
[1]Department of Physical Geography, Institute of Geographical Sciences, Freie Universität Berlin,
Malteserstr. 74-100, 12249 Berlin, Germany; subham.m@fu-berlin.de (SM); brigitta.schuett@fu-
berlin.de (BS)
[2]Chair of Landscape Planning and Development, Institute of Landscape Architecture and
Environmental Planning, Technische Universität Berlin, Straße des 17. Juni 145, Sekr. EB 5, 10623
Berlin, Germany; k.namba@tu-berlin.de (KN); stefan.heiland@tu-berlin.de (SH)
[3]Institute of Meteorology, Freie Universität Berlin, Carl-Heinrich-Becker-Weg 6–10, 12165 Berlin,
Germany; katrin.nissen@met.fu-berlin.de (KMN)
Correspondence to: Subham Mukherjee (subham.m@fu-berlin.de)
*Short summary*
*Berlin's parks are vital for recreation, biodiversity, and climate resilience, yet they face growing*
*challenges from socio-economic inequalities and climate change. Our review examines how factors like*
*gentrification and extreme weather impact access to and sustainability of these parks. By analysing*
*over 200 studies, we highlight the need for inclusive policies, community engagement, and climate-*
*adaptive park designs to ensure that Berlin's parks remain accessible, resilient, and socially just.*
**Abstract:**
Berlin, known for its rich history and lively cultural tapestry, boasts an extensive network of urban parks
that serve as vital lungs for its residents, providing recreational opportunities, ecological services, and
respite from urban life. These green spaces face multifaceted challenges from shifting socio-economic
dynamics and escalating impacts of climate change. This review article delves into the intricate interplay
between socio-economic conditions and the impact of climate change on Berlin's urban parks.
More than 200 research articles, reports, and policy papers on urban parks, green space management,
biodiversity, socio-economic challenges, and climate change are reviewed that explores how the
combined impact of socio-economic vulnerabilities and climate change intensifies the need for
sustainable, equitable, and resilient urban ecosystems. By adopting an intersectionality perspective, it
examines the complexities of these issues and reviews current management practices and policy
approaches. The review emphasizes the importance of inclusive green space planning, social
engagement, and targeted policy interventions to address these challenges.
Socio-economic disparities play a significant role in shaping unequal access to urban green spaces,
highlighting the broad relationship between social inequality and the use of these public resources. The
imbalances in access, quality, and affordability of these spaces, examining their implications for
different communities are explored. Gentrification, often driven by the appeal of green neighbourhoods,
raises the spectre of displacement and social exclusion, making the intersectionality of socio-economic
and environmental issues ever more pressing. Simultaneously, climate change poses new and escalating
threats to urban parks in Berlin, with rising temperatures, more frequent extreme weather events, and
biodiversity loss challenging these green oases. Case studies reveal innovative approaches, such as
community-driven transformations and climate-resilient park designs, that hold promise for achieving
sustainability.

*Keywords: Urban Green Spaces, Climate Resilience, Biodiversity, Environmental Justice, Community*
*Engagement*





## 1. Introduction:

Urban parks and greens are crucial elements of city life, contributing significantly to live-ability, environmental quality, and residents' well-being (Panagopoulos et al., 2016; Parker and Simpson, 2018). In Berlin, a city with dynamic urban development, these green spaces characterize cityscape and hold large importance (Lachmund, 2013; Kronenberg et al., 2020). This study investigates how climate change and climate extreme events impact *urban parks* in Berlin, considering varying socio-economic conditions, and, thus, aims to foster sustainable urban ecosystems. The review paper explores how socio-economic factors, climate change highlighting extreme weather impact Berlin's urban parks, emphasizing the growing challenges posed by more frequent and intense climate-driven events. The primary objective is to comprehensively understand the intricate socio-environmental dynamics at play within urban parks, more specifically, which are public spaces, as opposed to other types of greenery such as private gardens or roadside trees. These other types of greenery will also be considered when discussing general bio-physical and social interactions. This in-depth analysis, based on a systematic review of literature either as peer-reviewed journal articles or government documents, endeavors not merely to mitigate impacts, but to elucidate the complex interplay of ecological, social, and economic factors. Through this nuanced understanding, we seek to develop informed recommendations that will foster the creation and maintenance of sustainable urban ecosystems.

Berlin, known for its history, culture, and urban life, has a strong connection to greenery (Brantz and Dümpelmann, 2011). Understanding Berlin's urban parks, thus, requires a historical perspective (Angelo, 2021). In contemporary Berlin, urban parks serve purposes beyond just aesthetics and leisure (Li, 2023). Ongoing urbanization demands a re-evaluation of their role (Lehmann, 2012). For example, the transformation of Tempelhofer Feld from an airport into a community park and then (partly) a refugee-shelter exemplifies this shift (Owens, 2018).

Re-viewing sustainability for Berlin's urban parks from an intersecting society-ecosystem-policy perspective is a response to evolving climate and society. It emphasizes the interplay between ecological integrity, social equity, and economic viability within Berlin's green spaces (Ricci, 2022; Kotsila et al., 2023). This re-viewed sustainability encompasses unique ecosystem services (Fontaine, 2013), emphasizes inclusivity (Anguelovski et al., 2020), acknowledges economic benefits (Edwards, 2005), addresses climate resilience (Abbass et al., 2022), and calls for flexible and adaptive governance models (Renn and Klinke, 2013; Green et al., 2016). Despite challenges, such as in its traffic policies, Berlin's aspirations for sustainability and efforts to balance environmental responsibility, social equity, and economic goals offer valuable insights for advancing global green city initiatives (Alibašić, 2018; Ricci, 2022).

The concept of urban sustainability revolves around the capacity of cities to maintain or enhance the well-being of current and future urban residents while minimizing environmental impacts (Spiliotopoulou and Roseland, 2020; Sheikh and van Ameijde, 2022). This concept of multidimensionality serves as a central theme within the context of intersectionality, which is the primary focus of our paper. Intersectionality recognizes that individuals and communities possess multiple intersecting identities based on factors such as race, gender, class, age, and sexuality, which shape their experiences and access to resources (Davis, 2014; Lindley et al., 2021). Applying intersectionality to urban sustainability means acknowledging that sustainability challenges and benefits are not evenly distributed among all urban residents (Castán Broto and Neves Alves, 2018; Anguelovski et al., 2020). By critically assessing the literature, it becomes evident that this framework is essential for understanding the complexities of urban sustainability in a diverse city like Berlin.

The aim of this review is to analyse how socio-economic conditions and climate change impact on the social functions and sustainability of urban parks in Berlin. Specifically, it seeks to answer the following research question: *What are the scientific recommendations for sustainably maintaining and developing Berlin's urban parks to ensure their social functions and adaptability to climate change, considering the interlinkages between the two?* Additionally, the review examines whether these recommendations are reflected in the City of Berlin's current plans and strategies.

The review begins with a description of the methodology, detailing the systematic review process; it then presents an analysis of how socio-economic factors and climate change affect the ecological, social, and economic roles of urban parks. Finally, the discussion synthesizes these findings to propose recommendations for enhancing the sustainability and resilience of Berlin's green spaces in response to present and future challenges.



## 2. Study area: Berlin

Berlin, Germany's capital, presents a detailed case study for the development of its extensive urban
green network amidst a rapidly growing population (Figure 1). Spanning a city area of more than 89,000
hectares, Berlin's population is projected to grow significantly, with forecasts predicting approximately
4 million residents by 2040; this growth trend is expected to continue (Statistical Office Berlin-
Brandenburg, 2024). Additionally, Berlin hosts a substantial immigrant community, with over half a
million residents contributing to the city's demographic composition (Amt für Statistik Berlin-
Brandenburg, 2024).
Despite the notable population growth, Berlin maintains a substantial portion of its area as green spaces.
Over 30% of the city is covered by green spaces, including public parks, forests, private gardens,
allotment gardens, cemeteries, recreational areas, sports grounds, and street greenery (Kabisch and
Hasse, 2014). Specifically, public green spaces and forests cover around 5246 hectares of the city, which
is part of the total area designated as green (Kabisch and Hasse, 2014). However, while residential areas
have seen an 18% increase over the past decade, the expansion of green spaces has not kept pace,
highlighting the need for innovative integration of green spaces within the growing city (Amt für
Statistik Berlin-Brandenburg, 2024).

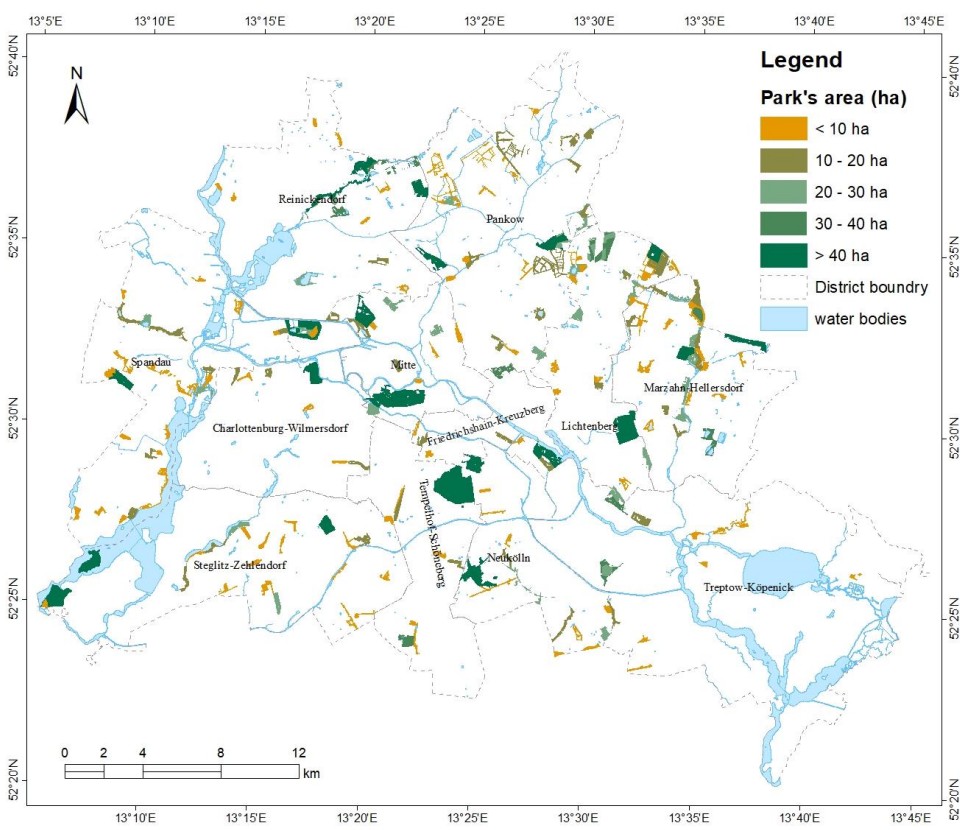

Figure 1. Map depicting the study area: Berlin city and its parks categorized by area, including water
bodies such as the River Spree (adopted from FIS Broker, https://fbinter.stadt-berlin.de/fb/index.jsp).

The evolution of Berlin's urban green spaces is deeply intertwined with the city's historical narrative,
reflecting its cultural, political, and social transformations. In the 19th century, landscape architects
such as Peter Joseph Lenné played a pivotal role in converting royal estates into public parks like





Tiergarten and Volkspark Friedrichshain. This transformation marked a shift towards recognizing the
importance of greenery in urban life, making these spaces accessible for public leisure and recreation
(Brantz and Dümpelmann, 2011; Wolschke-Bulmahn and Clark, 2021). In the 20th century, Berlin's
parks became arenas of political significance, mirroring Berlin's turbulent socio-political landscape.
Iconic spaces such as Tempelhofer Feld and Mauerpark today symbolize the city's division during the
Cold War and its later reunification, illustrating the complex role of green spaces in reflecting and
shaping societal changes (Angelo, 2021).
Concurrently, Berlin's urban parks are integral to the city's ecological, social, and economic fabric. They
contribute to biodiversity, mitigate the impacts of climate change, and serve as vital cultural and social
hubs, enhancing the well-being of its residents (Gandy, 2014; Kowarik, 2023). Economically, these
green spaces boost property values, attract tourism, and stimulate local economies, though this growth
can lead to challenges such as gentrification, which necessitates a careful balance between economic
development and social equity (Collins et al., 2022; Vargas-Hernández et al., 2023). Additionally, parks,
in general, have been crucial for public health, offering essential spaces for relaxation and physical
activity, particularly during the COVID-19 pandemic, underscoring their role in mental health and
community resilience (Collins et al., 2022).
However, Berlin's green spaces face significant challenges in ensuring ecological sustainability, social
inclusivity, and economic balance. The city's efforts to adapt to climate change, ensure equitable access
for all residents and manage economic disparities are critical to the future of these spaces (Stoetzer,
2018 and 2022; Amorim-Maia et al., 2023). The repurposing of former industrial sites, such as the
transformation of Görlitzer Bahnhof into Görlitzer Park in the late 1980ies or Berlin-Tempelhof Airport
into a vast urban park in the 2010s, exemplify the city's ongoing innovative approach to integrate green
spaces into its urban landscape (Draus et al., 2021). These efforts highlight Berlin's commitment to use
its green network as a tool to navigate the complex challenges posed by socio-economic shifts and
climate change (Kabisch and Hasse, 2014; Lachmund, 2013).
**3. Review approach:**
This review employs a systematic approach to identify, analyse, and synthesize relevant academic
literature on urban parks in Berlin. The focus is on understanding the intersectionality between Berlin's
evolving socio-economic conditions, climate change impacts, and the role of urban parks in fostering
sustainability. By adhering to established systematic review protocols, the methodology involves a
thorough, predefined search strategy, selection criteria, and critical evaluation process. This ensures a
robust and unbiased examination of literature that spans socio-environmental studies, historical
overviews, and case-specific investigations relevant to Berlin's urban parks.
The following key components are included:
**Socio-environmental Studies:** To understand the contemporary significance of urban parks in Berlin,
an analysis of existing research on socio-environmental studies have been conducted. These studies
involve the collection of academic literatures related to the ecological impact of these green spaces,
their cultural and social relevance, economic implications, and their role in enhancing residents' well-
being.
**Case-specific Investigations:** Further, case-specific literature survey on selected urban parks in Berlin
is included, that offer detailed insights into how those urban parks in Berlin have been shaped by the
city's history and continue to evolve in response to contemporary challenges. We investigated the
transformations and adaptations of these spaces through localized data collection and analysis.
Applying a systematic analytical approach includes a including a representative sample of research
articles were that address the intersectionality between Berlin's changing socio-economic conditions,
climate change impacts, and their influence on urban parks, with a focus on achieving sustainability.
An exhaustive keyword search was conducted across academic databases to across academic databases
to identify relevant articles, utilizing platforms such as PubMed, Scopus, Web of Science, and Google
Scholar. The following keywords and combinations were used:
- Berlin
- Urban parks
- Greenspaces
- Socio-economic conditions
- Climate change



- Sustainability
To be included in the review, academic papers had to meet the following criteria:
1) **Relevance:** Papers had to directly address the intersectionality of socio-economic conditions,
climate change impacts, and urban parks and greens, in general, and urban parks, in particular,
within the context of Berlin.
2) **Publication Type:** Only peer-reviewed journal articles and conference papers published in
English were considered.
3) **Publication Date:** A comprehensive literature review was conducted to encompass the
historical and contemporary understanding of urban green spaces and extreme weather
events. Scholarly articles and reports were included from across the entire available publication
spectrum, except for those specifically listed in the Appendix, till May 2024. This inclusive
approach ensures the analysis considers the full range of relevant research, providing a robust
foundation for understanding these critical issues.
Papers were excluded from consideration if they fell into any of the following categories:
1) **Non-English Language:** Papers published in languages other than English were generally
excluded due to limited translation resources. However, the study did include websites, reports,
and articles in German, as well as other non-academic materials from both governmental and
non-governmental organizations (after verification), to provide relevant examples. References
to these non-academic articles and reports are typically provided in the footnotes.
2) **Irrelevance:** Papers that did not directly address the intersectionality of socio-economic
conditions, climate change impacts, and urban parks and greens, in general, in Berlin were
excluded.
3) **Publication Type:** Books, theses, reports, and non-peer-reviewed articles were excluded to
maintain the academic rigor of the selection.
The initial search yielded a total of 634 academic papers. These papers underwent screening based on
title and abstract to exclude those not meeting the inclusion criteria. Following this screening, 308
papers remained for full-text review. Each of these papers underwent a critical assessment to evaluate
its relevance to the research topic.
After the full-text review, a final selection of around 200 academic publications was made based on
their direct relevance to the intersecting subject areas of Berlin's changing socio-economic conditions,
impacts of climate change, and urban parks within the context of sustainability. These selected papers
formed the foundation for the analysis and synthesis presented in this review article.
The final selection of papers covered a wide range of topics, methodologies, and findings, facilitating a
comprehensive and multifaceted exploration of the research area. Incorporating these papers ensures
that the review offers a well-rounded and informed perspective on the subject matter, integrating various
research approaches and insights to inform the discussion and conclusions of the article.
By amalgamating background analysis, socio-environmental studies, and case-specific investigations,
this review approach enables a comprehensive exploration of the complex relationships between
Berlin's urban parks, socio-economic conditions, and climate change. Moreover, it provides a robust
empirical foundation for the subsequent sections of this article, which delve into the multifaceted
challenges and opportunities faced by these green spaces in Berlin.
**4.   Synthesizing key Insights from Reviewed Literature**
The extensive literature search on Berlin's parks as sustainability infrastructure in the face of climate
change yielded a diverse array of academic papers. These papers (more than 200, altogether listed in
the *Reference* section) span multiple disciplines, time periods, and geographical focuses, offering a
comprehensive understanding of how urban green spaces in Berlin contribute to the city's resilience and
sustainability. This section provides a critical analysis of the selected papers, categorized by discipline,
year of publication, and focal study area, to contextualize their relevance within the broader discourse
on urban sustainability and climate adaptation.
**A. Disciplinary Breakdown**
The selected papers can be assigned to five primary disciplines (Figure 2): Urban Planning and Design,
Environmental Science and Ecology, Social Sciences and Urban Studies, Climate Science and
Meteorology, and Public Health and Well-being.




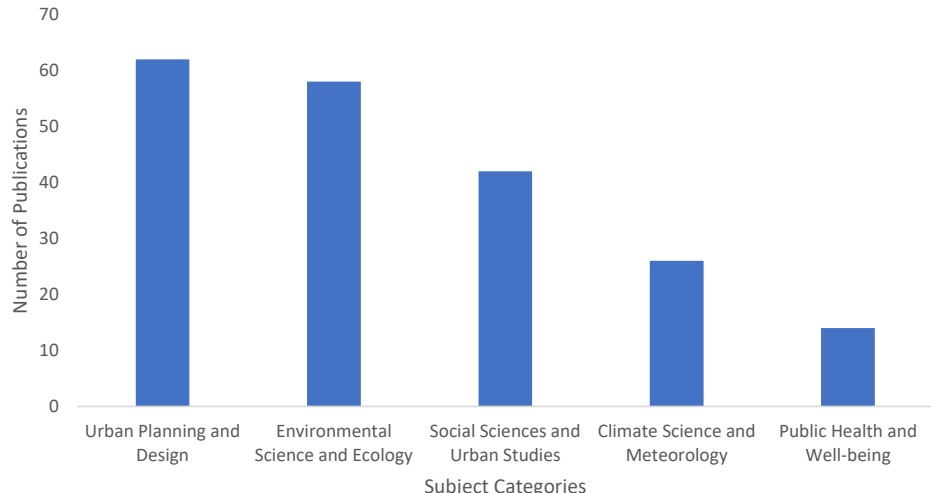

Figure 2. Disciplinary breakdown of the selected papers in the literature review.

a) **Urban Planning and Design:** This category comprises about 30.7% of the selected papers. The focus here is on the planning, design, and implementation of green spaces in urban settings, specifically how these spaces function as critical infrastructure within the urban fabric of Berlin. Key contributions from this discipline include discussions on the integration of green spaces into urban planning frameworks, the challenges of densification, and the role of parks in enhancing urban liveability (e.g., Lachmund, 2013).

b) **Environmental Science and Ecology:** Approximately 28.7% of the publications reviewed fall under this category. These studies primarily explore the ecological functions of urban green spaces, including biodiversity conservation, ecosystem services, and the role of green infrastructure in mitigating urban heat islands and managing stormwater. Berlin's parks are frequently examined as case studies for understanding urban biodiversity and the ecological benefits of green spaces in densely populated areas (e.g., Kowarik, 2023).

c) **Social Sciences and Urban Studies:** This category accounts for roughly 20.8% of the papers. The focus is on the socio-cultural implications of urban green spaces, such as their role in fostering social inclusion, mitigating gentrification, and promoting community well-being. The intersection of urban green space development with issues of social equity and justice is a recurring theme, particularly in studies examining the impacts of green gentrification in Berlin (e.g., Anguelovski et al., 2020).

d) **Climate Science and Meteorology:** Around 12.9% of the selected papers are from these disciplines. These studies are crucial in understanding the direct and indirect impacts of climate change on urban areas, with a specific focus on Berlin. Topics include the increasing frequency and intensity of extreme weather events, such as heatwaves and heavy rainfall, and the role of green spaces in mitigating these effects. The papers highlight how Berlin's green infrastructure can help the city adapt to changing climatic conditions (e.g., Fenner et al., 2019).

e) **Public Health and Well-being:** The remaining 6.9% of the papers focus on the health-related benefits of urban green spaces. These studies examine how access to parks and green areas contribute to physical and mental health, especially in the context of urban environments. In Berlin, the relationship between green space availability and public health outcomes is a key area of investigation, with several studies linking park accessibility to improved well-being during periods of extreme heat and other climate-related stressors (e.g., Kabisch et al., 2021).

**B. Year of Publication**





The papers reviewed span over a decade, with an increase in publications over the last five years (Figure
3). This temporal distribution reflects the growing importance of urban green spaces in climate
adaptation strategies and the rising academic interest in Berlin's response to climate change.

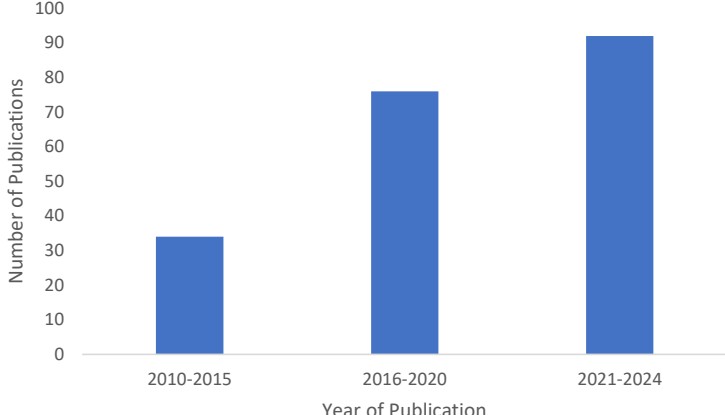

Figure 3. Temporal distribution of the reviewed papers where the bars show the total number of
publications during the time-interval mentioned.

a) **2010-2015:** During this period, about 16.8% of the reviewed papers were published. These early
studies primarily laid the groundwork for understanding the role of green spaces in urban planning
and environmental management in Berlin. Topics included initial explorations into green
infrastructure and its potential to enhance urban resilience (e.g., Wolch et al., 2014).
b) **2016-2020:** This period saw a significant increase in publications on the city's urban greens,
accounting for 37.6% of the publications, considered for review in this study. The focus shifted
towards the integration of green spaces into broader urban sustainability frameworks and addressing
the socio-political challenges associated with urban green space development, such as gentrification
and social equity (e.g., Bernt, 2016).
c) **2021-2024:** The most recent period accounts for 45.5% of the publications reviewed, reflecting the
heightened urgency in addressing climate change impacts on urban areas. The studies from this
period are particularly relevant to the current discourse on climate adaptation, exploring how
Berlin's parks are leveraged as key infrastructure to mitigate the impacts of extreme weather events,
such as heatwaves and heavy rainfall (e.g., Baganz and Baganz, 2023).
**C. Focal Study Area**
The focal study area of the selected papers primarily centres on Berlin, Germany, with some studies
including comparative analyses with other global cities (Figure 4). Berlin is a unique case study due to
its historical, political, and social context, making it an ideal subject for examining the intersection of
urban green spaces and sustainability.





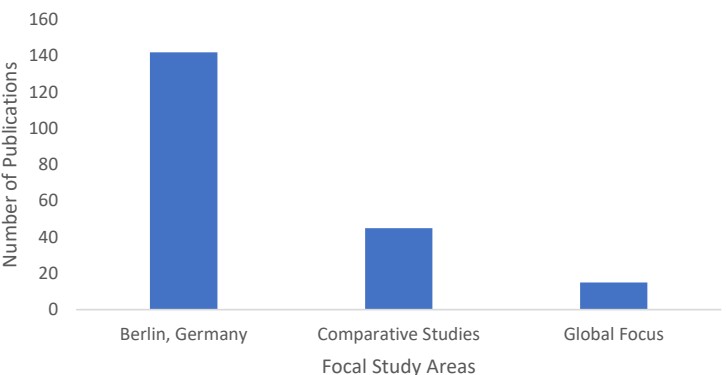

Figure 4. Geographical focus of the selected papers.

a) **Berlin, Germany:** Approximately 70.3% of the papers included in the review focus exclusively on Berlin. These studies explore a wide range of topics, from the ecological functions of parks to their role in social cohesion and climate adaptation. The emphasis on Berlin highlights the city's innovative approaches to urban green space management and its challenges in balancing development with environmental sustainability (e.g., Breuste and Breuste, 2022).

b) **Comparative Studies:** About 22.3% of the papers include Berlin as part of a comparative study with other cities, such as Leipzig, London, and New York. These studies provide valuable insights into how Berlin's green space strategies compare with those of other cities, offering lessons in good practices and highlighting areas where Berlin's approach can be improved (e.g., Ali et al., 2020).

c) **Global Focus:** 7.4% of the papers included in the review have a broad, global focus, but still reference Berlin as a case study within a wider context. These studies often discuss global trends in urban sustainability and climate resilience, positioning Berlin within the global discourse on how cities can adapt to and mitigate the effects of climate change (e.g., Gill et al., 2007).

## 5. Climate Change and Urban Parks: Impacts on Berlin's Biophysical Systems

Urban parks in Berlin, like their counterparts around the world, face a growing threat from climate change (Fryd et al., 2012; Jansson, 2013; Shade et al., 2020; Angelo, 2021). In Berlin a statistically significant temperature increase can be observed since 1950; the linear trend implies a rise of the annual mean temperature of 2.1°C (0.028 K/yr; adj. $R^2$:0.39) as well as of the annual minimum (4.8°C; trend: 0.07 K/yr; adj. $R^2$: 0.11) and maximum temperature (3.4°C; trend:0.046K/yr; adj. $R^2$: 0.21) (Figure 5). While annual mean precipitation does not show any statistically significant trends, the number of dry days has increased (23.4 d, trend:0.316 d/yr; adj. $R^2$: 0.09), indicating a shift towards lesser but extremer rainfall events. This shift is predicted to increase with rising greenhouse gas concentrations (e.g., Nissen et al. 2017).  The following subsections examine the impact of climate change on urban parks in Berlin exploring the implications of rising temperatures and extreme weather events.

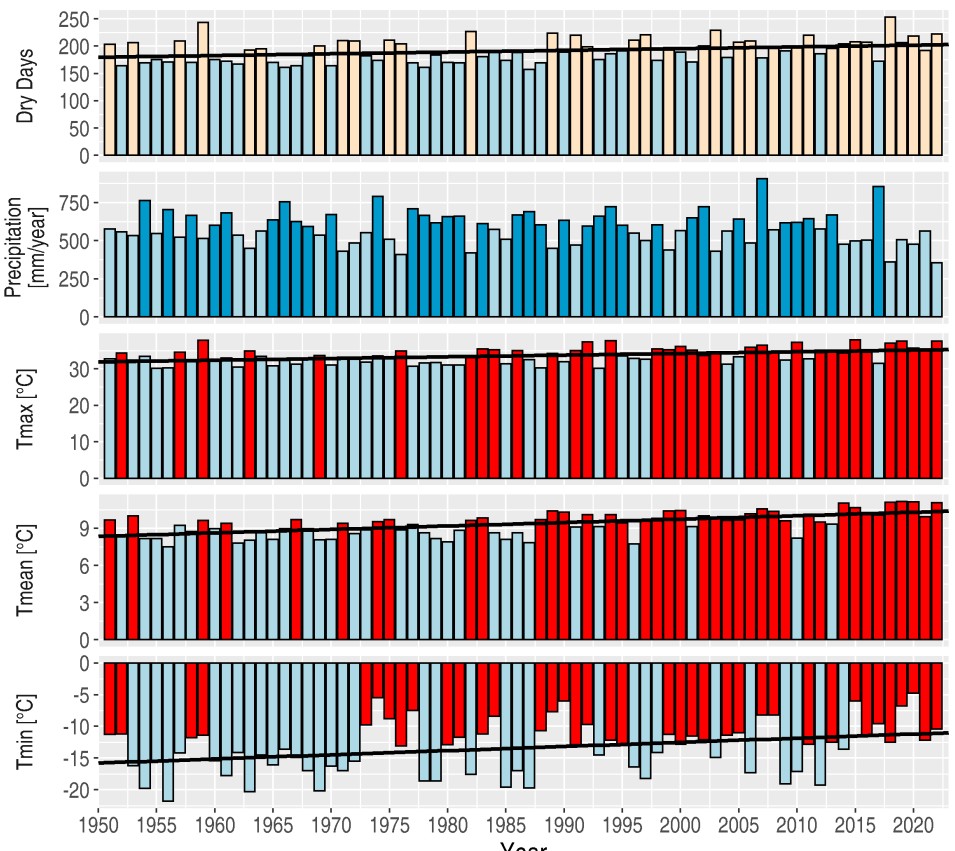

Figure 5: Climate Trends in Berlin (1950-2023): Precipitation and Temperature Variations with
Statistical Significance. From top to bottom: The number of days without precipitation per year
(beige/blue more/less than the long-term mean), Annual precipitation (light/dark blue less/more than
the long-term mean), absolute temperature maximum of the year (blue/red lower/higher than the long-
term mean), average temperature of the year (blue/red lower/higher than the long-term mean), and
absolute temperature minimum of the year (blue/red lower/higher than the long-term mean). The long-
term mean is based on the period 1950-2023. Black lines denote statistically significant linear trends
(5% level) (Data source: the weather station Berlin-Dahlem, Germany).
*5.1. Rising Temperatures: Urban Heat Islands effects*
Rising annual temperatures are a global phenomenon driven by climate change, and temperatures in
Berlin follows this trend (Abbass et al., 2022; Sander and Weißermel, 2023). However, the urban heat
island (UHI) effect, which occurs independently of climate change, also plays a significant role in
elevating temperatures in urban areas and aggravates the effects of climate change. UHI arises from
urban structures like concrete, asphalt, and buildings that absorb and radiate heat, making cities,
including their green spaces, warmer than surrounding rural areas (Marando et al., 2022). While climate
change exacerbates this effect, UHI would still exist in cities even without global warming, as it is
inherently linked to the urban form and density (Tsoka et al., 2020; Marando et al., 2022; Irfeey et al.,
2023). As a result, urban parks in Berlin experience heightened heat stress during the summer, impacting
both visitors and residents (Kabisch et al., 2021; Xu et al., 2022).
Climate Analytics conducted a study on heat stress and adaptation measure in Berlin and Brandenburg
commissioned by the Climate Change Centre Berlin Brandenburg. Their project report emphasizes the
importance of green spaces and sustainable urban planning to mitigate the effects of climate change,



particularly heat stress, in urban areas (Climate Analytics, 2024). Using the example of Greifswalder
Strasse in Berlin, the authors analysed various development options to enhance resilience for heat stress.
Their study suggests that a combination of reduced ground surface sealing and the creation of large,
contiguous green spaces (biotope networks) with trees is the most effective strategy for reducing heat
stress.
*5.2. Current State and Significance:*
**Implications for Park Functionality:** Thermal stress in Berlin during hot spells is lower in parks and
other green spaces compared to built-up areas, making them important cooling refuges (Langer et al.,
2020). However, while excessive heat primarily discourages people from leaving their homes, those
who do venture outside may still experience discomfort in parks, particularly if shade and water access
are limited (Kabisch et al., 2021; Lo et al., 2022; Xu et al., 2022). For vulnerable populations, such as
the elderly and young children, prolonged exposure to high temperatures—even in green spaces—can
pose health risks (Kabisch et al., 2021). This underscores the need for urban parks to be designed with
climate resilience in mind, ensuring they remain accessible, comfortable, and inclusive spaces for
recreation and well-being (Reyes-Riveros et al., 2021).
**Ecological Consequences:** Rising temperatures, both from climate change and the urban heat island
effect, have significant ecological implications for Berlin's urban parks and green spaces (Kraemer and
Kabisch, 2022; Kowarik, 2023). Some plant species may struggle to adapt to the warmer conditions,
leading to shifts in biodiversity, where certain species thrive while others dwindle (Lehmann, 2021).
However, such shifts are not inherently negative; urban biodiversity has historically been dynamic,
particularly in cities where alien species have contributed to increased species richness, a unique feature
of urban ecosystems (Kowarik and Ranger, 1994; Kowarik, 2019, 2023). Wildlife inhabiting the urban
spaces also faces challenges due to rising temperatures as changes in temperature can disrupt seasonal
behaviors, affecting breeding, migration, and feeding patterns of birds, insects, and mammals (Hsiung
et al., 2018; Kubelka et al., 2022). These disruptions may contribute to further shifts in biodiversity
(Koleček et al., 2020), but as with plants, urban wildlife has shown resilience, with new and non-native
species sometimes enriching the ecological fabric of cities (Kowarik, 2023; Stoetzer, 2022).
*5.3. Extreme Weather Events:*
Climate change brings a heightened risk of extreme weather events, including droughts, heavy rainfall,
storms, and flooding (Hettiarachchi et al., 2018, Caldas-Alvarez et al. 2022). Berlin's urban parks (and
greens, in general) are not exempt from these impacts (Fenner et al., 2019; Eckstein et al., 2021).
**Flooding:** Intense rainfall events can lead to pluvial flooding in urban parks, causing damage to
infrastructure (Alexander et al., 2019), eroding soil (Hazelton and Murphy, 2021), and potentially
affecting plant life (Czaja et al., 2020; Zipperer et al., 2020). Parks situated in low-lying areas are
particularly susceptible (Mehtab and Kamal, 2023). Flooding not only disrupts park activities but also
necessitates costly repairs and can pose safety hazards to visitors (Southon and van der Merwe, 2018).
**Damage to Park Infrastructure due to Natural Hazards:** According to the IPCC (2012), a hazard is
defined as the potential occurrence of a natural or human-induced physical event that may cause loss of
life, injury, or other health impacts, as well as damage and loss to property, infrastructure, livelihoods,
service provision, and environmental resources. In the context of urban parks, the specific hazard is
damage from extreme weather events, such as storms (Miller, 2020). Trees, pathways, recreational
facilities, and infrastructure within parks are particularly vulnerable to such damage. This vulnerability
can lead to temporary closures of parks, necessitate costly rehabilitation efforts, and pose safety risks
(Yildirim et al., 2021). The functional capacity of these spaces and the services they provide to the
community can be severely disrupted by storm-related damage (Karaye et al., 2019; Miller, 2020).
*5.4. Biodiversity Loss:*
Biodiversity is a fundamental component of urban park ecosystems, contributing to their resilience and
sustainability (Gonçalves et al., 2021; Lehmann, 2021). It includes the variety of plant species, the
presence of wildlife, and the intricate web of ecological relationships that develop in these green spaces
(Aerts et al., 2018; Heydari et al., 2020).
In the context of the climate change, biodiversity loss emerges as both a consequence and a
compounding factor of weather extreme events such as heatwaves, droughts and, flash floods due to
heavy rainfall, which overwhelm insufficient infrastructure like sewage systems. Climate change



intensifies these events, which can degrade habitats, reduce species populations, and disrupt ecological
balance, further accelerating biodiversity loss (Lehmann, 2021). While biodiversity loss is driven by
multiple causes—including habitat fragmentation, pollution, and invasive species—its role in the
climate crisis is particularly significant because reduced biodiversity diminishes urban parks' ability to
mitigate and recover from extreme events (Heydari et al., 2020). Therefore, addressing biodiversity loss
within the context of climate-driven extreme events is critical to understanding the broader impacts on
biophysical systems in urban parks.
**Species Migration:** Climate change influences the distribution of plant and animal species (Mashwani,
2020). As temperatures rise, some species may need to migrate to more suitable habitats, both within
and outside the city (Keeffe aandnd Han, 2019). In the context of Berlin's urban parks, this migration
can disrupt established ecological relationships (Stoetzer, 2018; Kowarik, 2023). The composition of
species in these green spaces may shift, impacting the balance and dynamics of these ecosystems
(Breuste et al., 2020; Baganz and Baganz, 2023).
**Vulnerability of Native Species:** Native plant and animal species within urban parks may face
increased competition from invasive species that are better adapted to the changing climate (Alizadeh
and Hitchmough, 2019). This competition for resources and habitat can lead to shifts in species
composition and a potential decline in the richness of native flora and fauna (Storch et al., 2022). The
loss of native species can have cascading effects on the overall functioning of the urban park ecosystem
(Carboni et al., 2021; Park and Razafindratsima, 2019). Ecosystem services are a vital aspect of urban
park functionality (Mexia et al., 2018). These services encompass a range of benefits provided by
ecosystems, including urban parks, that contribute to the well-being and quality of life of the city's
residents (Pukowiec-Kurda, 2022).
**Pollination:** Urban parks play a crucial role in supporting pollinators, such as bees and butterflies
(Ayers and Rehan, 2021; Dylewski et al., 2019). These insects are essential for the pollination of plants,
including many food crops (Requier et al., 2023). Climate change can disrupt the timing and availability
of flowering plants, impacting pollinators' foraging patterns (Bhatnagar et al., 2019; Gérard et al., 2020).
This disruption can ultimately affect the pollination of food crops within and beyond the city, potentially
leading to reduced agricultural yields and increased food prices (Marshman et al., 2019; Requier et al.,
439 2023).
**Pest Control:** Ecosystem services provided by urban parks include natural pest control (Qiu, 2019;
Sikorski et al., 2021). Predatory insects and birds that inhabit these green spaces help regulate pest
populations in nearby agricultural areas (Rocha and Fellowes, 2020). Climate change can alter the
distribution and behaviour of these species, potentially leading to increased pest problems in both urban
and rural environments (Qiu, 2019; Skendžić et al., 2021).
*5.5. Other Effects of Climate Change on Ecosystem:*
**Air and Water Purification:** Urban parks contribute to air and water purification by absorbing
pollutants and filtering water. They act as green lungs in the city, helping to improve air quality and
maintain water quality. Studies show that green spaces significantly reduce air pollution through
deposition on leaf surfaces and improve water management by promoting infiltration and reducing
surface runoff (Vieira et al., 2018). However, rising temperatures and altered precipitation patterns due
to climate change can affect the park's ability to provide these purification services. High temperatures
can lead to increased ozone formation, reducing air quality benefits (Xing and Brimblecombe, 2019).
Altered precipitation patterns can affect the park's ability to filter and purify water, potentially resulting
in contamination of local water sources (Kuhlemann et al., 2020).
**Climate Regulation:** Urban parks play a role in local climate regulation by providing shade, reducing
heat, and mitigating the urban heat island effect (Langer et al. 2020). However, climate change can
challenge the parks' capacities to provide these services effectively. Increased heatwaves can test the
parks' ability to offer cooling and relief to visitors, especially to vulnerable population groups. Without
proper adaptation measures, urban parks may become less effective in mitigating extreme temperatures,
leading to heat-related health issues (Gabriel and Endlicher, 2011: Scherer et al. 2013).
**Overall Ecological Stability:** The ecosystem services provided by urban parks contribute to the overall
ecological stability of the city. They support biodiversity, enhance resilience to environmental changes,
and foster a healthier urban environment. Parks in Berlin have been shown to host a variety of plant and
animal species, contributing to urban biodiversity (Palliwoda et al., 2017). However, climate change-





induced disruptions to these services can undermine the ecological stability of these green spaces, affecting both wildlife and human residents. Changes in temperature and precipitation patterns can alter the habitat conditions within parks, making them less suitable for certain species and reducing the overall biodiversity (Battisti et al., 2019).

## 6. Green Spaces, Governance, and Socio-economic Dynamics in Urban Park Management in Berlin Berlin's Urban Park management

The interplay between green spaces and urban park management provides a foundational understanding of how Berlin's urban infrastructure and planning strategies intersect with broader socio-economic dynamics. By contextualising these dimensions, this section establishes the relevance of green infrastructure policies and initiatives as critical enablers of equitable access and social inclusivity in urban park management. This approach bridges the gap between governance frameworks and socio-economic disparities, offering a comprehensive lens through which to examine Berlin's urban parks.

As the concept of urban green space covers multiple dimensions ranging from parks, community gardens, parking lots, buildings with green roofs and facades in urban areas, one needs to analyse policies at different levels of governance (EU, federal, state, municipal) affecting local green space development in Berlin. At the global level, the Berlin's Senate adopted the Berlin Urban Nature Pact in September 2024, an international initiative that aims to mobilize cities around the globe to protect and restore nature in urban areas. [1]

Urban green spaces could also offer effective nature-based solutions for sustainable urban drainage systems (SUD) in reducing stormwater flows and combined sewer overflows (CSOs) for urban water management in Berlin (Wild et al. 2024). Implementing the Sponge City Concept especially in urban areas and using rainwater from private roofs to water public green spaces are also promoted in Germany's National Water Strategy (2023).[2] At the municipal level, Berlin has introduced various policy incentives to promote water-sensitive or climate proof infrastructure. For example, the city's strategy to reduce flood risk is through decentralized rainwater harvesting.[3] Berliner Regenwasseragentur (Berlin's Rainwater Agency), an initiative of Berliner Wasser Betriebe (BWB) and of Senatsverwaltung für Mobilität, Verkehr, Klimaschutz und Umwelt (SenUVK) promotes decentralized rainwater harvesting projects by installing green rooftops on buildings, unsealed parking places for storm water management etc. Berlin also provides incentives for those who use rainwater for private houses and gardens (Wild et al. 2024)[4]. Berlin's vision to develop climate friendly urban green spaces are reflected in StEP Klima (2011) and the StEP Klima KONKRET (2016), a strategic spatial concept followed by the city's Urban Development Plan Climate 2.0, StEP Klima 2.0 (2022).

Berlin's urban landscape strategy (*Strategie Stadtlandschaft*), adopted by the Senate in 2011, focuses on the development and enhancement of the city's diverse green spaces. The focus of the strategy is on climate change and resource-efficient cities, demographic change and cultural diversity. The strategy supported programs such as urban tree campaign and the mixed forest program.[5]

In 2020, the Berlin's Senate established the Charter for Berlin's Urban Green "Charta für das Berliner Stadtgrün" in order to ensure that urban development is also green development and adapted the action program for Berlin's Urban Green 2030 "Handlungsprogramm für das Berliner Stadtgrün 2030" with concrete projects, measures and instruments.[67]

**Landschaftsprogramm:** The landscape program, including the species protection program (LaPro), is a strategic, city-wide planning instrument for integrative environmental precautions. It pursues the goal of integrating ecological concerns into urban development at a city-wide level.[8] Moreover, the Berlin's administration has been engaged with the issues of environmental justice in its districts since 2008, not

---

[1] https://www.berlin.de/rbmskzl/aktuelles/pressemitteilungen/2024/pressemitteilung.1481549.php

[2] https://www.bmuv.de/download/nationale-wasserstrategie-2023

[3] https://www.bwb.de/de/schwammstadt-berlin.php

[4] https://regenwasseragentur.berlin/massnahmen/regenwasser-sammeln-und-nutzen/

[5] https://www.berlin.de/sen/uvk/natur-und-gruen/landschaftsplanung/strategie-stadtlandschaft/

[6] https://www.berlin.de/sen/uvk/natur-und-gruen/charta-stadtgruen/

[7] https://www.berlin.de/sen/uvk/_assets/natur-gruen/charta-stadtgruen/charta.pdf?ts=1683531724

[8] https://www.berlin.de/sen/uvk/natur-und-gruen/landschaftsplanung/landschaftsprogramm/



only due to population growth in the city but also because of growing concerns for climate related
challenges (SenStadt and SenMVKU, 2023).
Furthermore, the initiative called "Volksentscheid Baum" has drafted the "BäumePlus-Gesetz" (Trees
Plus Act) for Berlin, which is intended to enshrine measures to make Berlin "weather-proof and heat-
proof" by 2035. According to the drafted law, Berliners would be allowed to plant trees and shrubs
themselves on streets. [9]
There are diverse forms of how urban spaces are managed. For example, GrünBerlin is a state-owned
public enterprise that implements Berlin's political guidelines, and which are accompanied by
corresponding public supervisory bodies (Grün Berlin: https://gruen-berlin.de/en/company/about-
gruen-berlin). GrünBerlin runs several of the major parks in Berlin and represents a case of private
organization and territorial governance of land (Colding et al. ,2013).
Kabisch (2015) identifies key challenges in Berlin's urban green governance, including (a) increasing
development pressure due to population growth and financial constraints on the municipal budget, (b)
loss of expertise, and (c) low awareness of green space benefits among various stakeholders due to
insufficient communication. Climate change is expected to further intensify these challenges. In
addition to these issues, Berlin's urban green spaces are often shaped by informal practices, such as
community-led initiatives, temporary land use, and adaptive greening efforts (Draus et al., 2020).
Berlin's urban green spaces, including community gardens, have been at the center of struggles between
local governments, which were often skeptical of civic engagement, and social movements advocating
for public access to green areas. These tensions became particularly visible in the early 1980s when the
first community gardens emerged in West Berlin (Rosol, 2010; Colding et al., 2013).
After reunification, the city had an abundance of unused urban spaces (*Brachen*). However, financial
constraints on the municipal budget limited green space development (Kabisch, 2015). The lack of
public funds also led to various forms of temporary land use (*Zwischennutzung*), where former
industrial areas were repurposed into cultural centers and informal green spaces. In response to these
budgetary challenges, local politicians began advocating for increased civic engagement in managing
green spaces (Rosol, 2010; Colding et al., 2013).
*6.1. Social Disparities*
Social disparities are a defining feature of urban life, including Berlin's urban life, and they have a direct
influence on the utilization of urban parks, in particular, and greenspaces, in general.
**Access to Green Space:** Income disparities can lead to unequal access to green spaces. Wealthier
neighbourhoods often have more well-maintained parks, whereas low-income areas may lack such
amenities. As a result, residents of economically disadvantaged areas may have limited access to these
essential recreational and restorative spaces, exacerbating social inequalities. In terms of accessibility,
there are strong disparities in green space provisions at household and individual levels in major German
cities (Wüstemann et al., 2017). Also, in the context of European urban areas, vulnerable and
unprivileged groups of residents receive below-average green cooling, while upper-income residents,
nationals and homeowners experience above-average cooling provision (Rocha et al., 2024),
corresponding to the findings for Berlin.
Berlin's *Umweltgerechtigkeitsatlas* (Environmental Justice Atlas) 2021/2022 identifies neighbourhoods
most affected by environmental stressors such as air pollution, noise, and limited access to green spaces.
In 2023, a guideline for promoting environmental justice in Berlin's neighbourhoods was developed
through a participatory process involving local representatives and experts from the Senate (SenStadt,
SenMVKU). Regarding green space provision, the neighbourhoods most negatively affected are
highlighted in the map shown in Figure 6[10].

---

[9] https://www.baumentscheid.de/klimaanpassungsgesetz
[10] https://climateanalytics.org/publications/hitzestress-und-anpassungsma%C3%9Fnahmen-in-der-metropolregion-berlin-brandenburg

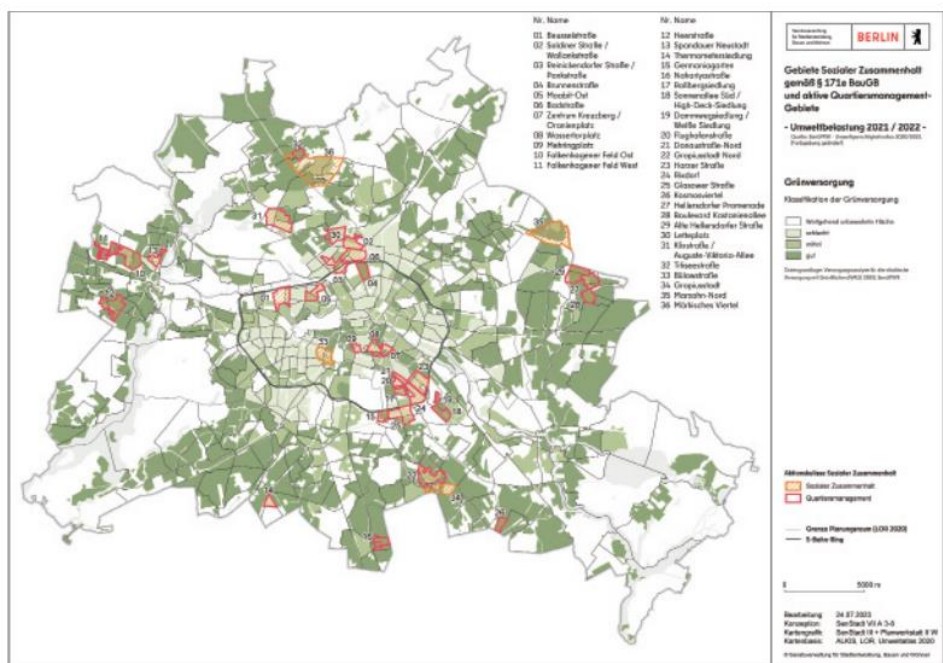

Figure 6: Provision of green spaces within the framework of social cohesion. (Source: Praxisleitfaden Umweltgerechtigkeit in Berliner Quartieren (SenMVKU, 2023))

Residents who suffer from multiple environmental burdens earn less money than the average income in Berlin. The Senate classifies the social status of the neighbourhoods such as Glasower Straße listed as "very low": around twelve percentage of people who live there are unemployed, and around 24 percent of the total population receive social security. 56% of children grow up in families that receive social security. [11][12]

**Affordability of Leisure Activities:** The ability to engage in recreational activities within Berlin's parks can be limited for individuals and families with low financial means due to costs associated with certain amenities and activities (Blokland and Vief, 2021). This economic barrier further reinforces the exclusion of low-income communities from the benefits of urban green spaces, particularly parks (Blokland and Vief, 2021). Research has shown that low-income families often face challenges such as a lack of well-maintained facilities, limited free recreational programs, and fewer organized activities in parks (Rigolon, 2016; Cohen et al., 2019). Additionally, economic disparities play a significant role in determining access to community recreation resources, which are essential for promoting physical activity and overall well-being (McKenzie et al., 2013; Blokland and Vief, 2021).

### 6.2. Gentrification and Displacement

Gentrification is a significant socio-economic phenomenon in Berlin, particularly in areas near urban parks. This process carries both opportunities and challenges for urban greens.

**Investment and Improvement:** Gentrification often brings increased investment in the neighbourhood, which can lead to park improvements, enhanced safety, and overall revitalization. This can make these spaces more attractive and accessible, thereby increasing their relevance in the urban fabric (Kabisch and Haase, 2014).

---

[11] https://climateanalytics.org/publications/hitzestress-und-anpassungsma%C3%9Fnahmen-in-der-metropolregion-berlin-brandenburg

[12] https://www.rbb24.de/politik/beitrag/2022/08/berlin-umwelt-gerechtigkeit-karte-kieze-laerm-hitze-luftverschmutzung.html





**Displacement and Social Exclusion:** On the flip side, gentrification can lead to the displacement of
long-standing, low-income residents. As high-income individuals and families move in, property values
and rents rise. Consequently, the communities that so far relied on these green spaces for social and
cultural activities may be pushed out, altering the socio-demographic makeup of neighbourhoods. This
displacement disrupts the social bonds that parks facilitate and can lead to social exclusion (Ali et al.,
2020). Moreover, the phenomenon of "green gentrification" highlights how improvements in green
spaces can inadvertently contribute to these processes of exclusion (Triguero-Mas et al., 2022).
**Informality and illicit activities:** In addition to the transformation of the abandoned railroad site into
Naturpark Südgelände, Berlin is also known for its green space, informality and illicit activities such as
criminalities (drug dealing or prostitution) in parks (e.g., Görlitzer Park) (Draus et al.,2020). A tension
emerged when former *Brachen* or wasteland spaces transferred from informal social gathering areas
into residential landscapes or public parks (Lachmund, 2003; Draus et al., 2020). The Berlin's city
administration selectively promotes some activities, such as those of 'urban pioneers' in Tempelhof and
turns a blind eye to others. For Tempelhof, this was a deliberate strategy, as those urban 'pioneers' were
mobilized by the city government to occupy the space with 'informal' activities such as urban
gardening. However, once the territory was 'settled', the net of control began to be extended. In this
context, the categories of formality and informality become confused (Draus et al., 2020).
*6.3. Access and Equity*
The concept of access and equity in relation to urban parks is central to understand their intersectionality
with socio-economic conditions.
**Inequitable Distribution:** The uneven distribution of parks, often favouring more affluent
neighbourhoods, results in an inequitable urban landscape. Low-income communities may have to
travel relatively long distances to access green spaces or contend with overcrowded parks, limiting their
ability to reap the associated physical, mental, and social benefits. Studies indicate significant disparities
in green space provision across German cities, with income being a major factor influencing access to
urban green spaces (Wüstemann et al., 2017). Additionally, the distribution of urban green spaces in
Berlin shows considerable dissimilarity by immigrant status and age, highlighting the need for equitable
planning (Kabisch and Haase, 2014).
**Social Inclusion:** Urban parks play a crucial role in fostering social inclusion, yet access varies
significantly among socio-economic groups. Wealthier communities often enjoy several opportunities
for social interaction, leisure activities, and cultural engagement within these spaces. In contrast,
socially vulnerable groups, including those with migration backgrounds and low-income levels, may
encounter social barriers that limit their participation and integration within urban park settings. This
disparity underscores the need for equitable access strategies to ensure that all residents can benefit
from the social advantages offered by urban greenspaces. The accessibility of urban green spaces can
significantly impact social inclusion, with disparities evident in who benefits from these spaces
(Wüstemann et al., 2017).
**Economic Resilience:** Socio-economic conditions directly affect the economic resilience of
communities living near urban parks. Gentrification can bring economic benefits, but it can also lead to
housing and social instability for displaced populations. Low-income communities may experience
gentrification as a threat rather than an opportunity, further accentuating income disparities. The
phenomenon of green gentrification, where park improvements lead to increased property values and
displacement of low-income residents, has been documented in various contexts, including Berlin (Ali
et al., 2020).
**7.   Examples of Sustainability Strategies Unveiled in Berlin's Parks**
This section critically explores strategies and approaches aimed at achieving sustainability within
Berlin's urban parks, considering the intersectionality of socio-economic conditions and climate change
impacts. It delves into innovative solutions and case studies that provide insights into how these
essential green spaces can evolve to meet the challenges of the 21st century.
*7.1. Equitable Access and Inclusion*
**Redistributive Green Space Planning:** Equitable access to urban parks requires a redistributive
approach to green space planning. It involves identifying areas with limited access to green spaces,



particularly in low-income neighbourhoods, and strategically locating or expanding parks to ensure
proximity and inclusivity. Additionally, considering residents' needs and preferences in the park design
process can foster a sense of ownership and inclusivity.
**Community Engagement:** Community engagement is a vital aspect of achieving equity and inclusion.
Involving local communities in park design and decision-making processes can lead to more tailored
and community-responsive green spaces. This approach enhances the sense of belonging and
encourages active participation in park activities (Kurth, 2022).
**Example 1- The "Tempelhofer Freiheit":** Tempelhofer Freiheit, the former Tempelhof Airport turned
urban park, exemplifies the potential of inclusive green space planning. Its adaptive reuse was guided
by community input and ensured that the park remains accessible to a diverse range of Berliners. The
park now hosts various recreational and cultural events, providing a model of community involvement
and inclusive design (Bartoli and Heyden, 2017; van Ham and Klimmek, 2017; Pegorer, 2023; Ranzato
and Broggini, 2023; Chen et al., 2021). Tempelhof also plays a crucial role as intersection between
formal and informal space (Draus et al. 2020). The Helmholtz Center of Environmental Research (UFZ)
conducted a study which concludes that the Tempelhofer Feld was a unique place for society and nature
(Brenck et al. 2021). Maintaining the Tempelhofer Feld is also contested. One perspective favour
preserving the parkland, while other political entities in Berlin advocate for developing at least some
portions of the area of the park[13] for housing.
*7.2. Resilience and Climate Adaptation*
**Resilient Park Design:** To address the impact of climate change, parks need to be designed keeping
resilience in mind. This involves implementing climate-adaptive features such as green infrastructure,
tree planting, and water management systems (Pancewicz, 2021). Creating shaded areas, installing
fountains, and incorporating natural elements can help to mitigate heat stress. In Berlin, parks can be
envisioned as interconnected green corridors but also facilitate wildlife movement and enhance
ecological resilience, even amidst the challenges of an already densely built-up city facing increasing
population pressures.
**Example 2- Gleisdreieck Park:** Gleisdreieck Park in Berlin is a key example of resilient park design.
It connects several neighbourhoods, addressing inequities in green space distribution and offering
accessible green areas for diverse communities. The park's landscape is specifically designed to absorb
heavy rainfall, reducing the risk of flash flooding in the area by enhancing local water management
systems. In addition to its climate-adaptive features, it serves as an urban oasis that supports recreational
activities and promotes biodiversity, while acting as an integral part of the city's green infrastructure
network (Csizmadia et al., 2017; Naumann et al., 2018; Zaykova, 2021; Ferrari, 2023).
*7.3. Promoting Sustainability Through Community Engagement*
**Environmental Education:** Community-based sustainability programmes within urban parks involve
the offer of environmental education and of awareness initiatives. These programs can educate residents
about the importance of urban biodiversity, sustainable land management, and climate change
resilience. Teaching people how they can contribute to park sustainability, such as through responsible
waste management or wildlife protection, fosters a sense of stewardship.
**Eco-friendly Events:** Parks can host eco-friendly events that promote sustainable practices, such as
zero-waste festivals or environmental workshops. Encouraging event organizers to adopt sustainable
policies, reduce resource consumption, and minimize waste generation aligns these spaces with broader
sustainability goals.
**Example 3 - Tiergarten Park:** The Tiergarten Park exemplifies sustainable community programming.
It offers educational opportunities for residents and visitors, including wildlife observation and
environmental education activities. The Park also hosts eco-friendly events that promote sustainability
and responsible resource management, aligning with the city's commitment to a greener future (Zefkili,
2011; Lachmund, 2013; Skandrani and Prévot, 2015; Feld, 2017).
*7.4. Inclusivity in Gentrification Strategies:*

---

[13] https://leute.tagesspiegel.de/neukoelln/macher/2021/08/04/181017/was-die-parteien-mit-dem-tempelhofer-feld-vorhaben/





**Affordable Housing Provisions:** To ensure inclusivity in gentrifying areas near urban parks, city
planners can implement affordable housing provisions (Sainburg, 2023). These policies aim to maintain
socio-economic diversity in neighbourhoods experiencing gentrification, ensuring that low-income
residents can remain in these communities.
**Community Benefits Agreements:** Collaborative agreements between developers, the city, and local
communities can stipulate those investments in gentrified areas, including park improvements, come
with community benefits (Rosen, 2023). These agreements can include the allocation of resources for
affordable housing, job opportunities, and accessible green spaces that prioritize the needs of existing
residents (Michels and Hindin, 2022).
**Example 4 - Hasenheide Park:** Hasenheide Park in Berlin's Neukölln district highlights the importance
of affordable housing provisions and community benefits agreements (CBAs) in addressing
gentrification (Skandrani and Prévot, 2015; Hardinghaus et al., 2021; Collins et al., 2022). Affordable
housing policies can maintain socio-economic diversity by enabling long-term residents to stay in
gentrifying neighborhoods near urban parks (Kabisch and Haase, 2014). CBAs between developers, the
city, and communities ensure investments in parks, like Hasenheide, also fund affordable housing, job
opportunities, and accessible green spaces, prioritizing the needs of existing residents (Rigolon et al.,
2020; Rigolon and Nemeth, 2020; Martens et al., 2022).
**8. Discussion of Findings: Urban Parks as Essential 'Third Places' in Berlin Amidst Socio-**
**Environmental Challenges from Heavy Rainfall Events**
Urban parks serve as quintessential "third places," offering informal public spaces where individuals
gather for leisure, social interaction, and respite from urban life (Oldenburg, 1989). In Berlin, these
parks hold particular significance, as they not only contribute to the city's ecological and cultural fabric
but also serve as social hubs that bridge the divides between its diverse populations (Jeffres et al., 2009;
Purnell, 2015). However, the function of parks as third places is increasingly compromised by the
intensifying impacts of climate change, especially extreme rainfall events. This discussion
comprehensively examines the challenges facing Berlin's urban parks, analysing both the
environmental and social dimensions of climate change, financial constraints, and inequality in green
space access. By integrating these perspectives, the following sections explore potential strategies to
enhance the resilience and inclusivity of urban parks in Berlin.
While considering a range of meteorological phenomena, including heat waves and droughts, which are
well-documented in literature, this review prioritizes heavy rainfall events due to their unique and
significant challenges specific to Berlin's parks. While urban parks provide vital ecosystem services
such as climate regulation, flood mitigation, and social well-being, studies like Pasternack et al. (2020)
show that extreme rainfall events can overwhelm urban infrastructure, including parks, leading to
significant disruptions. Caldas-Alvarez et al. (2022) demonstrate that heavy precipitation in Berlin, such
as the June 2017 event, caused substantial economic damage and strained local resources. Unlike other
meteorological events, heavy rainfall leads to immediate runoff issues, soil erosion, and infrastructure
degradation in parks, as highlighted by Lorenz et al. (2019), who observed storm intensification in
urbanized areas of Berlin. The unique interaction between urban environments and precipitation
patterns, leading to increased risks from flash floods (heavy rainfall that cannot be *managed* by surface
and sewage system), makes it imperative to prioritize research on rainfall impacts over other weather
phenomena, which have already been extensively studied in Berlin's parks (Haase and Kabisch, 2014;
Lorenz et al., 2019; Pasternack et al., 2020; Kabisch et al., 2021; Caldas-Alvarez et al., 2022).
*8.1. Heavy Rainfall and Biophysical Disruptions in Parks*
Berlin's parks are not immune to the escalating frequency and magnitude of heavy rainfall events due
to climate change, which imposes significant stress on their biophysical environments. Intense rainfall
leads to soil erosion, waterlogging, and increased surface runoff, all of which deteriorate the parks'
ecological functions. Soil erosion, in particular, severely impacts the ability of parks to support
vegetation, retain water, and provide natural habitats for urban biodiversity (Sarah et al., 2015). As
erosion strips away topsoil, the ability of parks to absorb water and facilitate groundwater recharge is
compromised, resulting in worsened flood risks and the degradation of green space quality (Kowarik,
743 2023).





Compaction from frequent foot traffic in popular parks, combined with insufficient vegetation cover,
exacerbates these effects by reducing infiltration rates, which intensifies the volume of surface water
runoff. This, in turn, not only threatens the ecological integrity of theaffected parks but also limits their
ability to function as refuges during extreme weather events, such as acting as cooling zones during
heatwaves or spaces for respite during periods of heavy rain (Pancewicz, 2021). These disruptions
underscore the pressing need for sustainable park design that incorporates climate-adaptive features,
particularly in managing water flow and preventing soil degradation (Gill et al., 2007).
*8.2. Social Implications of Heavy Rainfall in Third Places*
Beyond the biophysical impacts, heavy rainfall events also undermine the social functions of parks as
third places. Waterlogged fields, flooded pathways, and damaged infrastructure render parks unusable
for extended periods, limiting access to spaces crucial for community engagement, physical activity,
and social interaction (Tomczyk et al., 2016). This problem is compounded for vulnerable
populations—such as the elderly, low-income residents, and migrant communities—who rely heavily
on public parks for recreation and as gathering places, especially in dense urban areas where private
green spaces are limited (Kabisch and Haase, 2014).
Various studies document that climate-induced disruptions to park accessibility disproportionately
affect these communities, exacerbating social inequalities in cities (Anguelovski et al., 2020). For
instance, marginalized groups are more likely to live in areas with fewer high-quality parks, and when
heavy rain renders these spaces unusable, their options for outdoor leisure become further restricted
(Wüstemann et al., 2017). In this sense, climate change exacerbates not only environmental
vulnerabilities but also entrenched social inequities, reinforcing the need for inclusive urban green space
planning that addresses both environmental and social dimensions.
*8.3. The Ecological and Social Instabilty: A New Reality for Urban Parks*
Heavy rainfall directly challenges the ecological stability of Berlin's parks, which are essential for urban
biodiversity and ecosystem services. Climate-driven shifts in precipitation patterns have been shown to
alter species composition, with some plant species thriving while others decline due to water saturation
or soil nutrient loss (Kowarik, 2023). Such shifts impact the broader urban ecosystem, leading to a
reduction in biodiversity and the degradation of ecosystem services, including pollination and natural
pest control, which are vital for maintaining healthy park environments (Reynaert et al., 2020).
This ecological instability also diminishes the parks' ability to function as social spaces, which is critical
to their role as third places. Flooded and poorly maintained parks discourage their use for social
gatherings, thereby weakening community ties. Studies on urban sociology emphasize that parks, as
third places, are particularly important in fostering informal social interactions that contribute to social
cohesion (Oldenburg, 1989; Purnell, 2019). The more parks are subjected to environmental degradation,
the less they can fulfill this role, especially for socio-economically disadvantaged groups who have
fewer alternatives for outdoor recreation (Byrne, 2017).
*8.4. Redefining Urban Parks as Resilient Third Places: The Role of Adaptive Strategies*
The compounded effects of climate change and social inequities necessitate a rethinking of how Berlin's
parks can continue to function as third places under increasingly unpredictable environmental
conditions. One critical approach is the integration of adaptive water management systems, such as
Sustainable Urban Drainage Systems (SUDS), which mitigate the impacts of heavy rainfall by
controlling runoff and preventing soil erosion (Gill et al., 2007). These systems not only enhance the
ecological resilience of parks but also ensure that they remain accessible during extreme weather events,
safeguarding their role as social spaces (Masson-Delmotte et al., 2021).
In addition to biophysical solutions, there is a growing need for participatory planning processes that
involve local communities in park management and adaptation efforts. Community engagement fosters
a sense of ownership and ensures that park designs reflect the needs of diverse user groups, particularly
those most affected by climate change (Kurth, 2022). Inclusive Park planning that prioritizes climate
resilience can help sustain the multifunctionality of parks as both ecological assets and social hubs,
thereby enhancing their ability to act as third places even in the face of environmental challenges
(Haaland and van den Bosch, 2015).



### 8.5. Equity in Access: Addressing the Social Dimension of Climate Resilience

The uneven distribution of green spaces across Berlin's neighborhoods underscores the importance of redistributive green space planning as a strategy for fostering equity in access to parks. Ensuring that all residents—especially those from marginalized communities—have equal access to climate-resilient parks is essential for promoting social equity in the city (Kabisch and Haase, 2014). This can be achieved by targeting investments in green infrastructure toward underserved areas, which often experience the highest climate vulnerabilities (Jeffres et al., 2009; Purnell, 2019).

Equity in access must also be considered when designing adaptive features, such as shaded areas and rain shelters, which can help parks serve as refuges during extreme weather events (Pancewicz, 2021). Without intentional planning that addresses these disparities, the benefits of climate-resilient parks may disproportionately accrue to wealthier neighborhoods, further entrenching social divides.

### 8.6. Toward Sustainable and Inclusive Third Places

Berlin's urban parks are at a critical juncture, where their continued function as third places is threatened by the dual pressures of climate change and social inequities. Heavy rainfall events, in particular, pose significant risks to both the ecological health of these parks and their ability to serve as inclusive social spaces. Addressing these challenges requires a holistic approach that integrates climate-adaptive infrastructure with socially inclusive planning processes. By rethinking the design and management of parks to prioritize resilience and equity, Berlin can ensure that its green spaces remain accessible and functional as third places for all residents, even in an era of increasing environmental unpredictability.

### 9. **Conclusions:**

This review article critically examines the state of Berlin's urban parks, unveiling a tapestry of findings that underscore their multifaceted roles and challenges. It also sheds light on the contested concept of urban green parks and spaces. From their historical evolution to contemporary significance, these green spaces serve as ecological, cultural, and social anchors in the urban landscape. Strategies and policy incentives for the transition towards sustainable urban parks are available. This review highlights the critical interplay between socio-economic conditions and climate change in shaping the accessibility, functionality, and sustainability of urban parks in Berlin. The findings underscore the need for targeted strategies and policy interventions that address these interconnections, ensuring that urban parks can fulfil their social functions while adapting to the challenges posed by climate change. Ultimately, this research contributes to a deeper understanding of how to manage and develop urban green spaces in a way that promotes equity, resilience, and sustainability. Many aspects and initiatives listed already exist, such as various federal and state level guidelines for urban green spaces and green infrastructure (e.g., Berlin's Sponge City Initiative or Urban Development Plan Climate 2.0.) However, more needs be put into practice in terms of increasing climate resilience and community involvement.

The review-analysis provides a significant contribution by systematically categorizing and synthesizing a wide array of academic literature on Berlin's urban parks, specifically as sustainability infrastructure in the context of climate change and heavy rainfall. This review is particularly important as it draws from multiple disciplines—urban planning, environmental science, social sciences, climate science, and public health—highlighting the complex, interconnected roles that urban green spaces play in enhancing urban resilience. The disciplinary breakdown and temporal analysis reveal that while traditional focuses on urban planning and environmental science are crucial, new insights from social sciences and public health are gaining prominence, emphasizing the importance of equity and public well-being in climate adaptation strategies. Additionally, the study's emphasis on Berlin, coupled with comparative analyses, sheds light on the city's unique challenges and approaches, offering fresh perspectives on how urban green spaces can be leveraged to meet contemporary climate and social challenges. The findings significantly advance the discourse by advocating for a more integrated, interdisciplinary approach to urban sustainability, which is essential for developing more resilient, inclusive, and adaptive urban environments in the face of escalating climate pressures.

Several research projects have proven the multifunctionalities of urban green parks for climate mitigation, adaptation, and enhanced community engagement. However, they also reflect and exacerbate socio-economic disparities while grappling with the mounting pressures of climate change. The synthesis of key findings illuminates the complex interplay between socio-economic conditions and environmental challenges, emphasising the imperative of equitable access, resilient design, and



community engagement for sustainable park management. While urban parks are integral components
of green spaces, the focus on Berlin's parks is deliberate. They are the most accessible and frequently
used green spaces for residents, offering essential recreational, ecological, and social benefits.
Understanding the unique challenges and opportunities faced by Berlin's urban parks is crucial for
developing targeted, sustainable solutions that can be scaled to other green spaces in the city.
Achieving sustainability in Berlin's urban parks necessitates confronting a myriad of interconnected
research challenges. Quantifying the economic value of ecosystem services provided by parks is a
complex endeavour also due to informality and illicit activities, requiring sophisticated methodologies
that capture the full spectrum of benefits they offer. From air purification to pollination, these services
underpin the ecological resilience and human well-being associated with urban greenspaces.
Understanding the intricate relationships between biodiversity changes and ecosystem service provision
further underscores the need for interdisciplinary collaboration, bridging disciplines such as ecology
and social sciences. So far, various initiatives in Berlin have addressed some aspects of these challenges.
Investing in climate-resilient park design emerges as a critical imperative in the face of escalating
climate risks. Balancing the competing demands of budget constraints and community needs, planners
must prioritise strategies that enhance resilience while fostering inclusivity and accessibility.
Approaches like green infrastructure and sustainable landscaping, which integrate both traditional and
modern techniques, can help to mitigate the impacts of extreme weather events while improving the
aesthetic and functional value of urban parks. Moreover, proactive engagement with local communities
is essential for co-creating adaptive solutions that address their unique needs and vulnerabilities. The
City of Berlin has already been addressing these issues through various policies and projects. The
missing link is a comprehensive approach that integrates these efforts and expands on them to include
broader community engagement and long-term resilience planning.
Addressing disparities in park access and quality requires a multifaceted policy approach grounded in
principles of equity and social justice. Redistributing green spaces to underserved areas is a crucial step
towards ensuring that all residents have equitable access to the benefits of urban nature. However, this
attempt must be undertaken with sensitivity to the concerns and preferences of local stakeholders,
navigating potential conflicts and trade-offs along the way. Moreover, fostering active community
involvement in park management and design is essential for cultivating a sense of ownership and
stewardship among residents, empowering them to shape the future of their green spaces. While Berlin's
policies already recognise these issues, further steps can include enhancing community programs and
integrating social equity into all levels of urban green space planning.
Our study presents a timely and essential contribution, offering a comprehensive interdisciplinary
analysis of Berlin's urban parks. By integrating perspectives from ecology, sociology, and economics,
we address both current challenges and emerging issues. Our focus on the intersection of socio-
economic disparities and climate change impacts provides a holistic framework for understanding and
enhancing urban green spaces. This integrative approach is crucial for developing resilient, inclusive,
and sustainable urban ecosystems. Moreover, our research underscores the urgent need for empirical
data and community-driven solutions, aiming to fill gaps in current policies and propose actionable
strategies that can be implemented at local, national, and international levels.
To overcome the limitations of the current review article and advance towards a more empirical
evidence-based understanding of Berlin's urban parks, a focused research agenda is imperative.
Leveraging interdisciplinary methodologies and advanced geospatial technologies can provide insights
into the impacts of extreme weather events on park ecosystems and community well-being.
Longitudinal studies tracking changes in park utilisation and biodiversity over time can offer valuable
insights into the resilience and adaptive capacity of these spaces. Moreover, partnerships with local
stakeholders and citizen science initiatives can enhance data collection efforts, promoting knowledge
co-production and empowering communities to contribute to climate change mitigation and adaptation
efforts. Achieving sustainability in Berlin's urban parks requires a holistic approach that addresses the
complex interplay between socio-economic conditions, climate change impacts, and equitable access.
By investing in empirical research, innovative design interventions, and community engagement
strategies, cities can ensure that urban parks remain resilient, inclusive, and lively in the face of ongoing
environmental challenges.
**Author contribution**



SM conceptualized the study and developed the initial framework of the manuscript, with support from KN and KMN. Together, SM, KN, and KMN conducted the systematic review, synthesizing key findings and drafting the manuscript. ER contributed to the visualization of results by creating maps and providing analytical insights. SH and BS critically reviewed and refined the manuscript, ensuring coherence and academic rigor.

**Competing interests**

KMN is one of the members of the editorial board of the journal – NHESS.

**Acknowledgements**

We would like to express our sincere gratitude to the Einstein Research Unit Climate and Water under Change (CliWaC), a project generously funded by the Berlin Senate, for providing the financial support and collaborative platform that facilitated the research and writing of this review article. Additionally, our heartfelt thanks go to the Berlin University Alliances (BUA) for their invaluable contribution to the development and interdisciplinary approach of this project.

We would also like to acknowledge the professors, postdoctoral researchers, and PhD students whose insightful contributions and thought-provoking discussions have shaped the ideas and refined the arguments presented in this article. Their dedication to the topic and their constructive feedback were instrumental in advancing the research. In particular, we extend our gratitude to our colleague Dr. Lena Masch from the Otto-Suhr-Institut für Politikwissenschaft, Freie Universität Berlin, for her valuable input in shaping the ideas and sources for this article. We also sincerely thank Dr. Tim Moss from Humboldt-Universität zu Berlin for his insightful contributions, particularly in providing a historical and contextual background on Berlin.

Finally, we extend a special thanks to Dr. Tobias Otte from Freie Universität Berlin for his immense effort, leadership, and management skills in driving the success of the CliWaC projects. His unwavering motivation, guidance, and continuous support have been crucial to the progression of this research, and we deeply appreciate his commitment to the team and the overall project.

**Financial support**

This work was supported by The Einstein Research Unit Climate and Water under Change (CliWaC) and the Freie Universität Berlin (FUB) in the framework of the Open Access Publishing Program.

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
