# Peer review of "Review article: Re-viewing Berlin's Urban Parks from the Perspectives of Socio-Economic Inequality, Climate Resilience, and Sustainable Management"

_EGUsphere, 2025_

## Author Comment (AC1)

**Response to Reviewer 1**

We sincerely thank the reviewer for their insightful and constructive comments. We have carefully revised the manuscript to address each point, ensuring greater clarity, academic rigour, and alignment with the aims of the paper. Below we provide a detailed point-by-point response.

**Reviewer 1**

**Comment 1:** The aim of the study is not clearly stated (lines 99–100). Sustainability is a very generic term, and the paper should clarify that its focus is on sustainable maintenance of urban green spaces and their contribution to resilience to climate extremes rather than adaptability of green spaces to climate change. **Posponso:** The aim statement has been revised to emphasise resilience and sustainable management.

Response: The aim statement has been revised to emphasise resilience and sustainable management under climate extremes, rather than generic sustainability or broad adaptability.

Revised text (Introduction):

"The aim of this review is to examine how socio-economic conditions and climate-related extreme events shape the resilience and sustainable management of Berlin's urban parks. Specifically, it seeks to answer the following research question: What scientific recommendations exist for maintaining and developing Berlin's urban parks in ways that safeguard their social functions and enhance their resilience to climate extremes, while accounting for the interlinkages between ecological, social, and economic dimensions? Additionally, the review investigates whether these recommendations are reflected in the City of Berlin's current strategies and planning frameworks."

Comment 2: The last paragraph of the introduction (lines 105–109) does not mention the case study. Response: We added an explicit transition to frame Berlin as the case study and linked examples. Revised text (end of Introduction):

"The review begins with a description of the methodology, detailing the systematic review process; it then presents an analysis of how socio-economic factors and climate change affect the ecological, social, and economic roles of urban parks. To ground these analyses in a concrete setting, Berlin is examined as a detailed case study. The city's historically layered and socially diverse park system—ranging from iconic spaces such as Tiergarten and Volkspark Friedrichshain to more recent transformations like Tempelhofer Feld and Mauerpark—offers valuable insights into resilience, inequality, and sustainable management. Finally, the discussion synthesizes these findings to propose recommendations for enhancing the sustainability and resilience of Berlin's green spaces in response to present and future challenges."

**Comment 3:** Chapter 5 does not show impacts from climate change; Berlin has a stable continental climate, and rising temperatures may be due to urbanisation. UHI would exist even without global warming.

**Response:** The section was revised to clarify that **UHI is urbanisation-driven but exacerbated by climate change**, ensuring accurate attribution.

**Revised text (Chapter 5, Rising Temperatures):**

"Rising annual temperatures are a global phenomenon driven by climate change, and Berlin reflects this broader trend (Abbass et al., 2022; Sander and Weißermel, 2023). At the same time, the urban heat island (UHI) effect, which occurs independently of climate change, significantly elevates local temperatures and aggravates the perceived impacts of warming. UHI arises from urban structures such as concrete, asphalt, and dense building forms that absorb and radiate heat, making cities—including their parks and green spaces—warmer than surrounding rural areas (Marando et al., 2022). While climate change amplifies this effect, UHI would persist even in the absence of global warming, as it is inherently linked to urban morphology and density (Tsoka et al., 2020; Marando et al., 2022; Irfeey et al., 2023). The interaction of these drivers means that urban parks in Berlin are increasingly exposed to heightened heat stress during summer months, with consequences for both ecological functioning and human well-being (Kabisch et al., 2021; Xu et al., 2022).

Climate Analytics (2024) conducted a study on heat stress and adaptation measures in Berlin and Brandenburg, commissioned by the Climate Change Centre Berlin Brandenburg. Their project report highlights the critical role of green spaces and sustainable urban planning in mitigating the combined impacts of climate change and urban heat, with a particular emphasis on reducing exposure to heat stress in

densely built-up environments (Climate Analytics, 2024). Using the example of Greifswalder Strasse in Berlin, the authors analysed a range of development scenarios to evaluate resilience options for addressing heat stress. The study concludes that the most effective strategy involves a combination of reduced ground surface sealing and the establishment of large, contiguous biotope networks with tree cover, which together can substantially lower urban heat loads and strengthen ecological connectivity."

**Comment 4:** Chapter 5.4 (Biodiversity Loss) attributes effects too strongly to climate change; urbanisation plays a major role.

**Response:** We revised the section to reflect the **compounding role of urbanisation and climate extremes together**.

**Revised text (Chapter 5.4, opening paragraph):**

"Biodiversity is a fundamental component of urban park ecosystems, contributing to their resilience and sustainability (Gonçalves et al., 2021; Lehmann, 2021). It includes the variety of plant species, the presence of wildlife, and the intricate web of ecological relationships that develop in these green spaces (Aerts et al., 2018; Heydari et al., 2020). In Berlin, biodiversity loss emerges from the combined pressures of urbanisation and climate extremes. Habitat fragmentation, pollution, and the spread of invasive species are intensified by weather-related events such as heatwaves, droughts, and flash floods that overwhelm insufficient infrastructure like sewage systems. These processes interact to degrade habitats, reduce species populations, and disrupt ecological balance, further accelerating biodiversity decline (Lehmann, 2021). While biodiversity loss is driven by multiple causes, its significance in the climate crisis is amplified because reduced biodiversity diminishes urban parks' ability to mitigate and recover from extreme events (Heydari et al., 2020). Therefore, addressing biodiversity loss requires recognising the compounded role of both urban development and climate-driven stressors to understand the broader impacts on biophysical systems in urban parks.

**Species Migration:** Climate change influences the distribution of plant and animal species (Mashwani, 2020). As temperatures rise, some species may need to migrate to more suitable habitats, both within and outside the city (Keeffe and Han, 2019). In the context of Berlin's urban parks, this migration can disrupt established ecological relationships (Stoetzer, 2018; Kowarik, 2023). The composition of species in these green spaces may shift, impacting the balance and dynamics of these ecosystems (Breuste et al., 2020; Baganz and Baganz, 2023).

Vulnerability of Native Species: Native plant and animal species within urban parks may face increased competition from invasive species that are better adapted to warmer or more disturbed conditions (Alizadeh and Hitchmough, 2019). This competition for resources and habitat can lead to shifts in species composition and a potential decline in the richness of native flora and fauna (Storch et al., 2022). The loss of native species can have cascading effects on the overall functioning of the urban park ecosystem (Carboni et al., 2021; Park and Razafindratsima, 2019). Ecosystem services are a vital aspect of urban park functionality (Mexia et al., 2018). These services encompass a range of benefits provided by ecosystems, including urban parks, that contribute to the well-being and quality of life of the city's residents (Pukowiec-Kurda, 2022).

**Pollination:** Urban parks play a crucial role in supporting pollinators, such as bees and butterflies (Ayers and Rehan, 2021; Dylewski et al., 2019). These insects are essential for the pollination of plants, including many food crops (Requier et al., 2023). Climate change can disrupt the timing and availability of flowering plants, impacting pollinators' foraging patterns (Bhatnagar et al., 2019; Gérard et al., 2020). This disruption can ultimately affect the pollination of food crops within and beyond the city, potentially leading to reduced agricultural yields and increased food prices (Marshman et al., 2019; Requier et al., 2023).

**Pest Control:** Ecosystem services provided by urban parks include natural pest control (Qiu, 2019; Sikorski et al., 2021). Predatory insects and birds that inhabit these green spaces help regulate pest populations in nearby agricultural areas (Rocha and Fellowes, 2020). Climate change can alter the distribution and behaviour of these species, potentially leading to increased pest problems in both urban and rural environments (Qiu, 2019; Skendžić et al., 2021)."

**Comment 5:** Air and water purification is linked to lower precipitation, but Figure 5 does not show a significant change. The ozone problem is due to UHI and transportation pollution, not climate change.

**Response:** The section and it's heading were revised to emphasise **UHI** and transport as main drivers, with climate change as an amplifier.

**Revised text (Chapter 5, Air and Water Purification):**

"Air Quality and Water Regulation: Urban parks contribute to air and water purification by absorbing pollutants and filtering water. They act as green lungs in the city, helping to improve air quality and maintain water quality. Studies show that green spaces significantly reduce air pollution through deposition on leaf surfaces and improve water management by promoting infiltration and reducing surface runoff (Vieira et al., 2018). In Berlin, however, the effectiveness of these services is shaped more by local urban conditions than by long-term climatic trends. Elevated ozone levels, for instance, are largely linked to transportation emissions and the urban heat island (UHI) effect, which intensifies pollutant concentrations during warm periods (Xing and Brimblecombe, 2019). Climate change can exacerbate these stresses by prolonging heatwaves, but it is not the primary cause. Likewise, while Figure 5 does not indicate a significant long-term reduction in precipitation, localised heavy rainfall events combined with extensive surface sealing can overwhelm park infrastructure, affecting infiltration and water purification capacity (Kuhlemann et al., 2020)."

Comment 6: The title of Chapter 6 repeats "Berlin's Urban Park management."

Response: The title has been clarified.

**Revised title:**

"Green Spaces, Governance, and Socio-economic Dynamics in Urban Park Management in Berlin"

**Comment 7:** Chapter 6 should discuss all urban green spaces (green infrastructure) and not only parks. **Response:** The opening of Chapter 6 was revised to acknowledge Berlin's broader green infrastructure, while still emphasising parks as the focal point of this review. A bridging section was also added after the Charter for Berlin's Urban Green to highlight legal frameworks.

**Revised text (Chapter 6, opening):**

"The interplay between urban green spaces and park management provides a foundational understanding of how Berlin's urban infrastructure and planning strategies intersect with broader socio-economic dynamics. By contextualising these dimensions, this section establishes the relevance of green infrastructure policies and initiatives as critical enablers of equitable access and social inclusivity in the governance of urban nature. This approach bridges the gap between governance frameworks and socio-economic disparities, offering a comprehensive lens through which to examine Berlin's green infrastructure, with particular emphasis on public parks as the most multifunctional and socially significant spaces. The concept of urban green space covers multiple dimensions ranging from parks, community gardens, allotment colonies, cemeteries, and urban forests to buildings with green roofs and facades. Accordingly, policies must be analysed at different levels of governance (EU, federal, state, municipal) that influence the development and management of local green spaces in Berlin. At the global level, the Berlin Senate adopted the Berlin Urban Nature Pact in September 2024, an international initiative that aims to mobilise cities around the world to protect and restore nature in urban areas."

**Bridging text (after Charter for Berlin's Urban Green):**

"Although Berlin's legal and strategic frameworks—such as the Public Parks Law (1997), the Charter for Urban Green (2020), and the Urban Green 2030 Programme—apply to the city's entire green infrastructure, parks remain their principal focus. These policies highlight the dual challenge of safeguarding ecological functions and ensuring equitable access, underscoring the centrality of parks in shaping Berlin's green future."

**Comment 8:** Conclusion is too extensive; last two paragraphs can be deleted.

**Response:** The conclusion has been restructured to avoid repetition, clarify the article's scope, and present findings in a more **concise**, synthesised manner. It now explicitly states that the article foregrounds parks within Berlin's broader green infrastructure.

**Revised text (Conclusion):**

"This review article critically examines the state of **Berlin's urban parks**, situating them within the city's wider green infrastructure but treating parks as the primary lens of analysis. The deliberate focus on parks reflects their prominence as the most multifunctional, accessible, and socially significant form of urban greenery in Berlin. While community gardens, allotments, green roofs, and other green spaces contribute to the city's resilience, this article analyses **parks in particular** to understand how socio-economic dynamics, governance challenges, and climate-related stressors converge.

Our findings highlight the interplay between socio-economic conditions and climate change in shaping the accessibility, functionality, and resilience of Berlin's parks. Strategies and policy incentives exist—such as the Sponge City Initiative, the Urban Development Plan Climate 2.0, and the Charter for Urban Green—but gaps remain in implementation. More systematic integration of resilience measures and stronger community involvement are needed to translate these frameworks into practice. Equity challenges are also persistent: income and social status influence access and quality, with disadvantaged groups disproportionately exposed to environmental burdens. These disparities underscore the urgency of embedding environmental justice principles into urban park governance.

By systematically synthesising literature across urban planning, environmental science, climate research, social sciences, and public health, this review provides a comprehensive interdisciplinary analysis. While ecological and planning perspectives remain foundational, emerging insights from social sciences and public health demonstrate the importance of equity, health, and well-being in resilience planning. Berlin's case illustrates both opportunities and tensions in managing parks to balance biodiversity conservation, climate adaptation, and social justice

Several research projects confirm the multifunctionality of Berlin's parks for climate mitigation, adaptation, and community well-being. Yet they also demonstrate that parks can reflect and exacerbate socio-economic disparities, particularly through processes of gentrification and uneven access. This synthesis highlights three interlinked priorities for sustainable management: enhancing resilience through climate-adaptive design, reducing socio-spatial inequalities in access and quality, and strengthening civic engagement in planning and stewardship.

Finally, the review points to a forward-looking research agenda. Quantifying the economic and ecological value of ecosystem services remains a complex challenge, requiring advanced interdisciplinary methods. Longitudinal studies are needed to trace biodiversity change, ecosystem services, and park utilisation over time. Partnerships with communities, supported by citizen science and co-produced knowledge, can enrich data collection while fostering stewardship. Future research should therefore integrate robust empirical evidence, inclusive governance, and adaptive design to ensure that Berlin's urban parks remain resilient, equitable, and vibrant in the face of escalating climate and social pressures."

Additionally, we have added a **closing sentence for Section 6** that neatly ties together Reviewer 1's request (broaden scope) with our article's main emphasis (parks) at the end Section 6:

"In sum, while Berlin's governance frameworks and policies address the entire spectrum of urban green infrastructure, this review foregrounds public parks as a key entry point for analysis. Parks remain the most multifunctional and socially significant spaces, making them particularly well suited for examining the intersections of socio-economic dynamics, governance challenges, and climate resilience."

We believe these revisions significantly strengthen the paper and thank the reviewers for their valuable feedback.

---

## Author Comment (AC2)

**Response to Reviewer 2**

We sincerely thank the reviewer 2 for their insightful and constructive comments. We have carefully revised the manuscript to address each point, ensuring greater clarity, academic rigour, and alignment with the aims of the paper. Below we provide a detailed point-by-point response.

**Reviewer 2**

**Comment 1:** *Include legal interventions (e.g., Public Parks Law 1997).*

**Response:** Incorporated into Chapter 6.

Bridging text (after Charter for Berlin's Urban Green):

"Although Berlin's legal and strategic frameworks—such as the Public Parks Law (1997), the Charter for Urban Green (2020), and the Urban Green 2030 Programme—apply to the city's entire green infrastructure, parks remain their principal focus. These policies highlight the dual challenge of safeguarding ecological functions and ensuring equitable access, underscoring the centrality of parks in shaping Berlin's green future."

**Comment 2:** *Maps and diagrams could be higher resolution.*

**Response:** Figures 1, 5, and 6 have been re-exported at high resolution (300–600 DPI) and captions updated for clarity. Additionally, photos associated with the case study parks are added with relevant captions, for better visual understanding.

**Comment 3:** Abstract could be more concise.

**Response:** Abstract shortened to 204 words, focusing on scope, methods, and key findings.

**Revised abstract:**

"Berlin, renowned for its rich history and vibrant cultural tapestry, possesses an extensive network of urban parks that function as vital lungs for the city, providing recreation, ecological services, and respite from urban life. Yet, these green spaces confront mounting pressures from shifting socio-economic dynamics and escalating climate-related impacts. This review investigates the intricate interplay between socio-economic conditions and climate change in shaping the resilience, accessibility, and sustainability of Berlin's parks. Drawing on more than 200 research articles, reports, and policy papers, it synthesises insights on park management, biodiversity, governance, and socio-economic disparities, with particular attention to their intersectionality. The findings highlight that socio-economic inequalities strongly influence patterns of access, quality, and affordability of green spaces, exposing disadvantaged communities to uneven benefits and environmental burdens. Processes of gentrification, often intensified by the appeal of green neighbourhoods, exacerbate displacement and exclusion, underscoring the need to integrate social justice into green space planning. Simultaneously, climate change introduces new threats, including rising temperatures, extreme weather events, and biodiversity loss, which compound urban vulnerabilities. Case studies from Berlin illustrate innovative strategies—ranging from community-driven initiatives to climateresilient park design—that demonstrate pathways towards inclusive, adaptive, and sustainable management of urban parks in the face of complex socio-environmental challenges."

We believe these revisions significantly strengthen the paper and thank the reviewers for their valuable feedback.